

# Observationally constrained surface mass balance of Larsen C Ice Shelf, Antarctica

Peter Kuipers Munneke[1], Daniel McGrath[2,3], Brooke Medley[4], Adrian Luckman[5], Suzanne Bevan[5], Bernd Kulessa[5], Daniela Jansen[6], Adam Booth[7], Paul Smeets[1], Bryn Hubbard[8], David Ashmore[8], Michiel Van den Broeke[1], Heidi Sevestre[9], Konrad Steffen[10], Andrew Shepherd[7], and Noel Gourmelen[11]

[1]Institute for Marine and Atmospheric research, Utrecht University, Utrecht, The Netherlands
[2]Colorado State University, Fort Collins, CO, United States
[3]US Geological Survey, Alaska Science Center, Anchorage, AK, United States
[4]Cryospheric Sciences Laboratory, NASA Goddard Space Flight Center, Greenbelt, MD, United States
[5]Geography Department, College of Science, Swansea University, Swansea, United Kingdom
[6]Alfred Wegener Institut, Bremerhaven, Germany
[7]School of Earth and Environment, University of Leeds, Leeds, United Kingdom
[8]Centre for Glaciology, Department of Geography and Earth Sciences, Aberystwyth University, Aberystwyth, United Kingdom
[9]Department of Geography and Sustainable Development, University of St Andrews, St Andrews, United Kingdom
[10]Swiss Federal Research Institute WSL, Birmensdorf, Switzerland
[11]School of Geosciences, University of Edinburgh, Edinburgh, United Kingdom

*Correspondence to:* Peter Kuipers Munneke (p.kuipersmunneke@uu.nl)

**Abstract.** Combining several geophysical techniques, we reconstruct spatial and temporal patterns of surface mass balance (SMB) over Larsen C Ice Shelf (LCIS), Antarctic Peninsula. Continuous time series of snow height at five locations allow for multi-year estimates of seasonal and annual SMB over LCIS. There is high interannual variability, with an SMB of $395 \pm 61$ to $413 \pm 42$ mm w.e. y$^{-1}$ in the north and a larger SMB of up to $496 \pm 50$ mm w.e. y$^{-1}$ farther south. This difference

between north and south is corroborated by winter snow accumulation derived from an airborne radar survey from 2009, which showed an average snow thickness of 0.95 m north of 76°S, and 1.12 m south of 78°. Analysis of ground-penetrating radar from several field campaigns allows for a longer-term perspective of spatial SMB: a particularly strong and coherent reflection horizon below 25–44 m w.e. of ice and firn is observed in radargrams collected across the shelf. We propose that this horizon was formed in a single melt season over the ice shelf. Combining ground and airborne radar with SMB output from a regional

climate model confirms that SMB increases from north to south, overprinted by a gradient of increasing SMB to the west. Previous observations show a strong decrease in firn air content toward the west, which we attribute to spatial patterns of melt, refreezing, and densification, rather than SMB.

## 1  Introduction

About 74% of the grounded ice sheet of Antarctica is drained to the Southern Ocean through floating ice shelves (Bindschadler

et al., 2011). By buttressing the grounded ice sheet (e.g. Dupont and Alley, 2005; Gagliardini et al., 2010), ice shelves strongly



modulate the flux of ice into the ocean, thereby exerting an important control over the contribution of mass variations of the Antarctic Ice Sheet to global sea level.

In recent decades, the collapse of ice shelves along the Antarctic Peninsula (Cook and Vaughan, 2010) was immediately followed by a sustained velocity increase of the glaciers previously feeding these ice shelves, by a factor 3–4 in documented

cases (De Angelis and Skvarça, 2003; Scambos et al., 2004; Berthier et al., 2012). It is likely that the current (and growing) mass imbalance of the Antarctic Peninsula (e.g. Shepherd et al., 2012; Harig and Simons, 2015) is due in part to these ice-dynamical adjustments to the loss of ice shelves. It is believed that at least part of the ice-shelf collapses along the Antarctic Peninsula can be related to warming of the near-surface atmosphere (Morris and Vaughan, 2003). Warming has been faster in this region than the global average since the 1950s (Marshall et al., 2006; Turner et al., 2014), in part caused by a very large

regional decadal variability (Turner et al., 2016). It has been hypothesized that enhanced meltwater production at the ice-shelf surface can lead to hydrofracturing, whereby meltwater-filled crevasses open up under the pressure exerted at the crevasse tip by the column of standing meltwater (Scambos et al., 2003; Van der Veen, 2007), and/or where drainage of meltwater lakes induces fracture by strong flexural stresses (MacAyeal and Sergienko, 2013). Kuipers Munneke et al. (2014b) suggested that the conditions for ponding and hydrofracturing depend on the local accumulation and melt fluxes, and their effect on the vertical

structure of the firn layer.

The Larsen C Ice Shelf (LCIS), the largest ice shelf of the Antarctic Peninsula, is located to the east of the north-south oriented Antarctic Peninsula mountain range (Figure 1). As the dominant upper-air wind direction is westerly, LCIS is in the climatological leeside of the mountains. This position gives rise to particular patterns in surface melt, with more melt and the occurrence of meltwater ponding in the inlets directly east of the mountains, and gradually less melt away from the mountains

towards the calving front in the Weddell Sea to the east (Trusel et al., 2013; Barrand et al., 2013). The advection of warm, dry air masses over the ice shelf during föhn winds is the likely cause of this surface melt distribution (Luckman et al., 2014). A notable expression of this surface melt gradient is seen in the composition of the firn layer over LCIS: the smallest amounts of firn air are found in the inlets in the western part of LCIS (Holland et al., 2011; Ashmore et al., 2016), and in Cabinet Inlet (Figure 1), the firn features a massive subsurface ice layer (Hubbard et al., 2016), influencing ice-shelf temperatures, density,

and potentially also flow properties.

However, although important, surface melt is only a part of the surface mass balance (SMB, see Data and Methods section, and Equations1 and 2). In addition to the observed spatial variability in surface melt, there are spatially varying patterns of snowfall and other SMB components that amplify or counteract the effect of surface melt on the observed gradients of firn air. Given that the SMB plays an important role in overall LCIS mass balance and controls its firn properties, the dearth of published

in-situ SMB over LCIS is problematic. In the most comprehensive compilation of Antarctic Peninsula SMB measurements to date (Turner et al., 2002), none were available on the ice shelves themselves. However, four observational records from sites on the grounded ice adjacent to LCIS are available. Two firn cores were collected on Dolleman Island (70.58°S, 60.92°W, 398 m a.s.l.), yielding mean SMB rates of 390 mm w.e. $y^{-1}$ for 1962–1982 (Mulvaney and Wolff, 1993), and 404 mm w.e. $y^{-1}$ (Peel and Clausen, 1982). A firn core on Gipps Ice Rise (68.77°S, 60.93°W, 290 m a.s.l.) gives 349 mm w.e. $y^{-1}$ (Peel and

Clausen, 1982). Stake observations by Rott et al. (1998) revealed a mean SMB of 360 mm w.e. on Jason Peninsula, which



forms the northern boundary of LCIS (66.25°S, 61.00°W). Based on an interpolation between these records, Turner et al. (2002) estimated annual SMB at <500 mm w.e. y$^{-1}$ over LCIS. However, given the prominence of orographic gradients in this region, one must be wary of extrapolating data from these elevated sites: high-resolution climate modelling as well as radar observations show that the SMB field around such features can deviate by up to ±50% relative to that of the flat surrounding terrain (Lenaerts et al., 2014). Finally, a re-analysis for the period 1979–1993 by Turner et al. (1998a) indicates that solid precipitation at a central point on LCIS was 490 mm w.e. y$^{-1}$.

Accumulation is much lower on the eastern side of the Antarctic Peninsula than on its western side. The latter is dominated by slow-moving low pressure systems over the Bellingshausen Sea, leading to a mostly northwesterly flow of humid and relatively mild air (Turner et al., 2002). Precipitation is orographically driven by the Antarctic Peninsula mountains. The largest precipitation events are associated with advection of moist air from mid-latitudes at times when a strong low-pressure system develops over the Bellingshausen Sea. In contrast, a continental climate exists on the eastern side. Barrier flow along the orography of the Antarctic Peninsula, resulting from a climatological low-pressure area over the Weddell Sea, leads to predominantly southerly flow over the eastern part of the peninsula (Parish, 1983). Precipitation events over LCIS are frequent and generally small (Turner et al., 1998a), and associated with (1) lee cyclogenesis over LCIS and the Weddell Sea immediately east of LCIS, and (2) active fronts arriving from the northeast or east (Turner et al., 1998b).

Over the past decade, several field campaigns have been undertaken to collect data on firn, SMB, meteorology, and climate over LCIS. During various field campaigns in 2008, 2009, 2010, 2011, 2014, and 2015, data have been collected using a number of geophysical techniques, including ground-penetrating radar, shallow firn coring, snow pits, and continuous snow height observations. Moreover, airborne radar data have been collected over LCIS in late 2009 and 2010.

The aim of this paper is to bring together this suite of data sets, in order to provide a coherent picture of SMB over LCIS, focusing on both the intra-annual, annual, and decadal time scales. These data can be used to evaluate the performance of atmospheric models over LCIS (King et al., 2015), such as RACMO2 (Van Wessem et al., 2016), AMPS (Powers et al., 2012), or CESM (Lenaerts et al., 2016). Models assessing ice-shelf stability require an estimate of SMB as a boundary condition (e.g., DeConto and Pollard, 2016), and their performance is subject to a correct representation of current melt and SMB over the ice shelf.

## 2 Data and Methods

### 2.1 Surface mass balance

The specific surface mass balance (in meter water equivalent per year, m w.e. y$^{-1}$) is defined as

$$SMB = \int\limits_{year} dt(PR - SU - RU - ER_{ds} - SU_{ds}), \tag{1}$$





where $PR$ is precipitation, $SU$ is surface sublimation, $RU$ is meltwater runoff, and $ER_{ds}$ and $SU_{ds}$ represent erosion and sublimation of drifting snow particles, respectively. Runoff is defined as

$$RU = ME - RF, \tag{2}$$

i.e. the difference between surface melt, $ME$, and internal refreezing within the snow and firn, $RF$. Strictly speaking, SMB, as used in this study refers to the mass balance of the entire firn column, not only of the surface.

Following Turner et al. (2002), we add the definition of mass in an annual layer (MAL) of the snow cover:

$$MAL = SMB - m. \tag{3}$$

Here, $m$ is the loss from an annual layer to lower or higher layers due to vapor transport or meltwater percolation. If $m$ is assumed to be small, MAL can be used as an estimate for SMB.

## 2.2 Sonic height rangers

Since 2009, sonic height rangers at various automatic weather stations (AWS) have been observing snow height (see Table 1 and Figure 1), yielding records of 3 to 7 years in length. A correction to account for the dependence of sound velocity on temperature is applied, using concurrent observations of air temperature. Sometimes, sonic pulses are reflected from the AWS mast or the datalogger box attached to the mast, giving erroneously low snow height readings. These observations are filtered out, and the resulting data gaps are filled by linear interpolation. Snow height observations are corroborated by manual measurements of the distance between the sonic ranger and the snow surface upon each annual maintenance visit. We assume an uncertainty in the surface height ranger readings of 0.10 m, which mainly represents noise due to small-scale surface roughness, as the accuracy of individual measurements is of the order of 0.01 m.

At all locations, sonic height rangers were attached to an AWS mast. Between 2009 and 2011, additional height rangers were placed at AWS 14 and 15 (see map in Figure 1), mounted at about 2 m above the surface on a separate triaxial construction consisting of three lightweight aluminium poles that were drilled into the snow at an angle of about $30°$, up to a depth of about 2.5–3.0 m below the surface. After 2011, we make use of the sonic height rangers attached to the AWS itself. We present a continuous snow surface time series that accounts for the occasional servicing of the SHR to avoid burial. In reality, the sonic height rangers were always located between 1 and 4 m above the surface.

All SHR data series exhibit data gaps, but unambiguous values for summer, winter, and annual surface height change could be derived for all years and all stations, except for one: the LAR1 time series is interrupted from 16 October to 9 November 2013. We have filled this gap with the mean elevation change of the other four sonic rangers, allowing us to compute a summer and winter surface height change at LAR1 for 2013.

## 2.3 Snowpits and firn cores

During various field campaigns between 2008 and 2015, snowpits have been dug to a depth of about 2 m below the surface (see map in Figure 1 for locations). In these snowpits, detailed snow stratigraphy was logged, and vertical profiles of density were



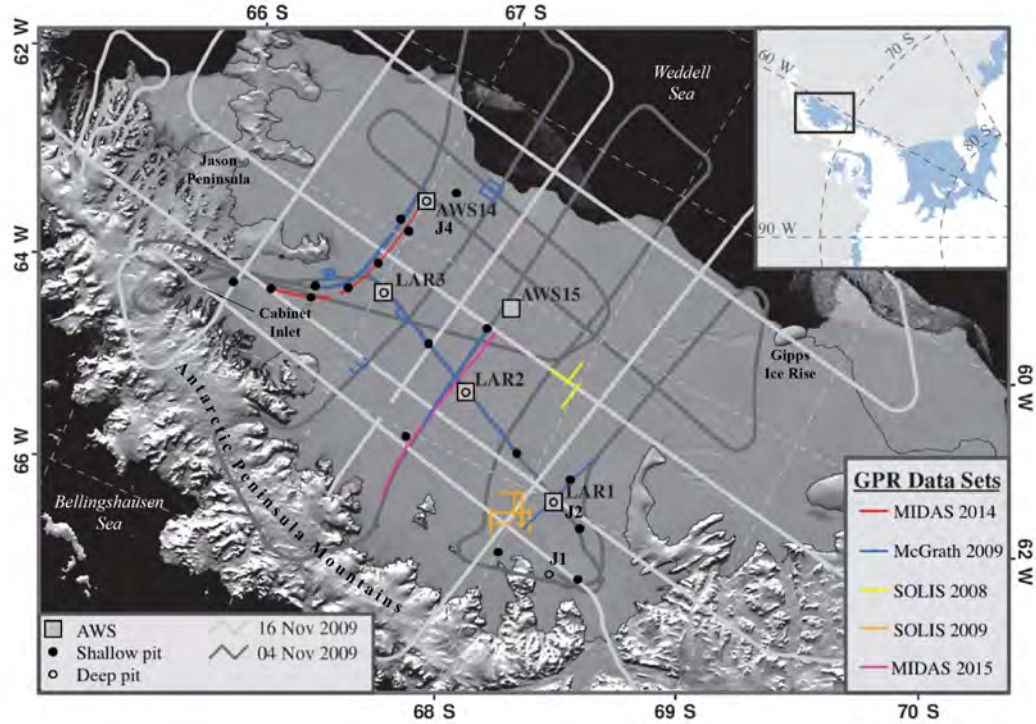

**Figure 1.** Map showing fieldwork locations, Operation Ice Bridge flight lines, GPR tracks, and relevant geographical names. The MODIS mosaic of Antarctica is shown in the background.

**Table 1.** Overview of sonic height rangers operated over LCIS and used in this study. Locations are shown on the map in Figure 1.

| Name | Lat (°S) | Lon (°W) | Start date | End date |
|------|---------|---------|-----------|----------|
| AWS 14 | 67.01 | 61.48 | Jan 2009 | still operational |
| AWS 15 | 67.57 | 62.13 | Jan 2009 | Jun 2014 |
| LAR1 | 68.14 | 63.95 | Dec 2008 | Apr 2015 |
| LAR2 | 67.58 | 63.26 | Dec 2008 | Nov 2011 |
| LAR3 | 67.03 | 62.65 | Aug 2009 | Nov 2011 |

recorded using a variety of tools and methods. In total, 22 shallow snow pits were used for the analysis in this paper. From these 22 snow pits — all collected before melt onset — we collected vertical profiles of density, resampled into 5-cm bins. We assign an uncertainty to the density observations of 20 kg m$^{-3}$, which is based on observations presented below.



In addition, three vertical profiles of density up to 11 m depth were collected using a neutron-scattering probe in 2009, at locations J1, J2, and J4, indicated on the map in Figure 1. In 2015, a 90-meter borehole was drilled with hot water, and surveyed with an optical televiewer (Hubbard et al., 2008) at the site of AWS14 (referred to as CI-120 in Ashmore et al., 2016). Using the vertical profile of luminosity as a proxy for density (Hubbard et al., 2013), an estimate of firn density is available for the entire length of the borehole.

All of these snowpits and firn cores provide an estimate of the density of the uppermost layers of firn to convert from radar two-way travel time to thickness, and from actual layer thickness to surface mass balance in water-equivalent thickness.

## 2.4 Ground penetrating radar

Between 2008 and 2015, five ground-based radar surveys have been carried out in different locations covering northern and central portions of LCIS (Figure 1) (Luckman et al., 2012; McGrath et al., 2012; Jansen et al., 2013; Kulessa et al., 2014; McGrath et al., 2014). Surveys were carried out using antennas of different frequencies (25, 100, or 200 MHz), depending on the field campaign. Despite the different frequencies, a distinct, spatially continuous reflector was identified in a large portion of the ground surveys. Several assumptions are necessary in order to convert the measured two-way travel time to this reflector ($\tau$) to snow accumulation rates.

First, we use an empirical relation between firn density to its dielectric constant Kovacs et al. ($\epsilon$ 1995), which determines the effective wave velocity through the firn. Thus, depth ($d$) is calculated as follows:

$$d = 0.5c\tau\epsilon^{-0.5}, \tag{4}$$

where $c$ is the speed of light in a vacuum. As $\epsilon$ depends on the firn density profile, and the accumulation rate (proportional to $d$) affects the density profile, an iterative technique is required to solve for $d$ in a consistent manner. The firn density profile can exhibit strong spatial variations due to differing surface melt rates and internal refreezing, snow accumulation rates, and air temperatures; however, field observations on Larsen C are sparse. Indirect measurements do exist, however, such as the map of firn air content (FAC) over LCIS derived by Holland et al. (2011), which provides insight into the spatial variations in firn density. This gridded product represents the firn air content as a column thickness, which requires a firn densification model in order to derive a density profile. Thus, we have modified a scheme developed by Medley et al. (2015) to solve iteratively for a density profile that is consistent with the Holland et al. (2011) FAC and the radar-derived accumulation rates. The method relies on the Herron and Langway Jr. (1980) semi-empirical densification model to produce steady-state depth-density profiles. Starting with an initial accumulation rate estimate and a constant initial density and long-term average temperature from the atmospheric model RACMO2 (see section below), we model the depth-density profile using Herron and Langway Jr. (1980). Accumulation rates are then derived from the radar horizons using that initial density profile, yielding a new long-term accumulation value. This new accumulation rate then replaces our initial estimate and the process repeats until convergence. This results in radar-derived accumulation rates that are consistent with the prescribed density data (i.e., the density profile is dependent on the accumulation rate, so it is necessary to ensure they are mutually consistent).





Because we have additional independent information in the form of the FAC, we use an additional iterative scheme to ensure the modeled FAC matches the Holland et al. (2011) gridded FAC. To accomplish this fitting scheme, we iterate the assumed initial density to reach the desired FAC. In a sense, we are lumping all changes in the FAC into the surface density, which does not necessarily represent a real surface process. In reality, surface melt is seasonal and meltwater infiltration and refreezing does not happen right at the surface, resulting in complex ice structures that can be interspersed with firn pockets throughout the vertical firn column (Ashmore et al., 2016). Therefore, in areas of high melt, the scheme will yield unrealistically high surface density estimates in order to accommodate for refreezing within the firn column that is not accounted for in our dry snow densification model. A modeling effort to include the complexity of meltwater infiltration, retention, and refreezing is beyond the scope of this work, and while our scheme is necessarily simplified, it provides reasonable bounds on the snow accumulation over LCIS sufficient to meet the needs of this work. Using our method, we generate radar-derived accumulation rates that are physically related to the modeled density profile, which is consistent with the FAC data of Holland et al. (2011).

To convert the estimated reflector depth to mass, we assume that the reflector is below the pore close-off depth, and then subtract the FAC thickness Holland et al. (2011) from the reflector depth, yielding an ice-equivalent thickness, which is converted to mass using an ice density of $910\,\mathrm{kg\,m^{-3}}$. Column mass is corrected for differences in acquisition dates of the various radar lines, assuming a mean accumulation of $450\,\mathrm{mm\ w.e.\ y^{-1}}$ in recent years.

## 2.5 Airborne snow radar and radar picking

Beginning in 2009, the NASA Operation IceBridge (OIB) campaign has annually surveyed both the Greenland and Antarctic ice sheets using a variety of instruments designed to map the geometry and internal structure of the ice. Two frequency-modulated continuous-wave (FMCW) radar systems, developed by the Center for Remote Sensing of Ice Sheets (CReSIS) at the University of Kansas, are capable of imaging the near surface stratigraphy of the firn column, hereafter termed the snow and Ku-band radars. For the 2009 campaign, the snow and Ku-band radars operated over the 4–6 GHz and 14–16 GHz frequency ranges, respectively; thus, the systems have the same bandwidth (2 GHz) and the same theoretical range resolution (∼6 cm in snow). While accumulation studies have exclusively used the snow radar up to this point, we use the Ku-band radar data because the snow radar was not operational during one of the two flights covering LCIS in 2009.

The Ku-band radar data set consists of two comprehensive surveys of the entire extent of LCIS on 4 and 16 November 2009, and one smaller survey along the western inlets on 31 October 2009. Therefore, the data were collected prior to the onset of surface melt. Prior work with the snow radar (e.g., Medley et al., 2013; Koenig et al., 2016) showed that strong, continuous radar reflections are observed over areas of West Antarctica and Greenland over hundreds of kilometers and to approximately 30 meters depth. By assuming that the layers are annually spaced, the radar stratigraphy provided spatially varying time series of snow accumulation. However, over LCIS we find a single, strong reflection horizon about 1 meter below the surface (see sample radar echogram in Figure 2). This horizon is the only apparent feature in both the snow and Ku-band radar data sets. Because the radar is sensitive to changes in density, we interpret the reflection as the the surface at the end of the previous melt season earlier that year, and by measuring its depth below the surface we can determine the amount of winter snow accumulation.





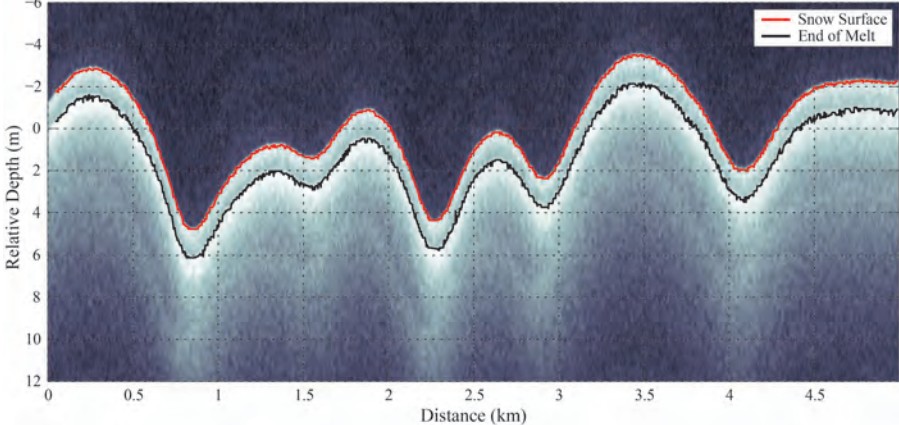

**Figure 2.** Sample radar echogram from the 16 November 2009 flight with automated surface and subsurface picks overlaid. Nearly all echograms look similar with very strong surface and a strong to very strong subsurface reflection horizon. The subsurface horizon is interpreted as the end of the melt season, and it physically represents the transition from dry snow (above) and snow that has strong melt features (ice lenses).

The radar data were not stacked because the strength of the reflection was well above the noise, making it easier to develop a simple, automated picking scheme. In an initial step, the surface reflection is picked automatically, and the radar two-way travel time for each trace is zeroed to the $\tau$ of the surface pick. Thus, $\tau$ is the two-way travel time relative to the surface. The picking scheme finds the next strongest reflection below the surface, which in this case is likely generated by the presence of ice lenses and metamorphosed firn created during the prior melt season. The subsurface picks are then filtered to exclude any extreme outliers using a running median filter. Finally, we inspect the result visually to ensure the automated picks are consistent with the visible stratigraphy. The two-way travel time to the subsurface reflection is converted to depth using a snow density of 360 kg m$^{-3}$ for the winter accumulation, that is derived from *in situ* snow pit observations (see section above). Again, we use the Kovacs et al. (1995) formula (Equation 4) to relate firn density to $\epsilon$, which determines the effective wave velocity through the firn. We calculate the depth uncertainty by combining the dominating uncertainty due to the picking scheme ($\pm 2$ range bins) with that of the density assumption ($\pm 20$ kg m$^{-3}$). Crossover analysis of the OIB radar-derived depths showed close agreement with an RMSE just under 2 range bins, which provides the basis for our uncertainty estimate due to layer picking.

## 2.6 Regional atmospheric climate model

RACMO2 is a regional climate model, optimized for simulation of climate and surface mass balance in polar regions. RACMO2 contains dedicated parameterizations for e.g. drifting snow and snow albedo. In this study, we use data from a RACMO2 simulation in a domain covering the Antarctic Peninsula and surrounding polar ocean at a horizontal resolution of 5.5 x 5.5 km, covering the period 1979-2014 (Van Wessem et al., 2016).





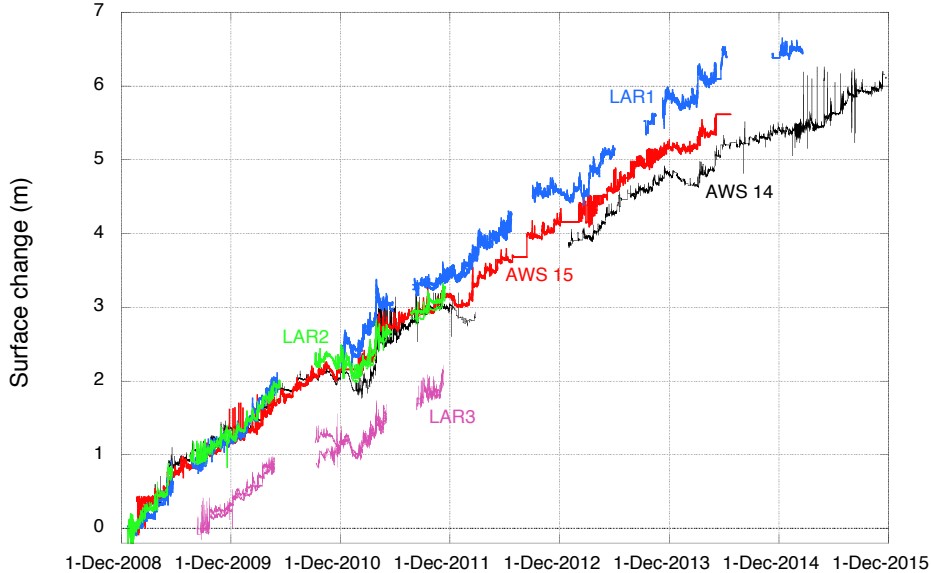

**Figure 3.** Complete, filtered time series of cumulative surface height change (m) recorded with sonic height rangers at five locations. LAR1, 2, and 3 were equipped with two sonic rangers, plotted in the same color.

## 3 Results

### 3.1 SMB estimates from sonic height rangers

We present the complete time series of snow height for the five sonic rangers at LCIS (Figure 3). At all locations, the surface height increased quite gradually, suggesting that precipitation occurs in a large number of small events, rather than in a small number of large accumulation events per year. In the summer months, a surface decrease is observed at all stations, consistent with expected melting of the surface snow, and refreezing in the firn below, and with enhanced firn compaction at elevated temperatures.

The multi-year records from sonic height rangers can be used to establish in-situ estimates of annual and seasonal SMB, provided that a reliable estimate of snow and firn density is available to convert surface elevation increase to mass accumulation. First, we estimate the density of the winter accumulation, i.e. before the start of the melt season, from the 22 snow pits investigated (see Methods). We excluded the part of the density profile below melt layers that likely originated from the previous melt season. In that way, we obtained the density of the winter accumulation. The 22 pits show no discernable temporal or spatial variability in the density of the winter accumulation. We therefore computed one mean vertical profile, shown along with the individual density profiles in Figure 4a. The vertically-averaged (+/- one standard deviation) density of the winter accumulation is $360 \pm 20$ kg m$^{-3}$. Combining this value with wintertime (Apr-Oct) sonic height ranger observations, we can construct annual time series of wintertime SMB. Figure 5 shows that interannual variability of wintertime SMB is large





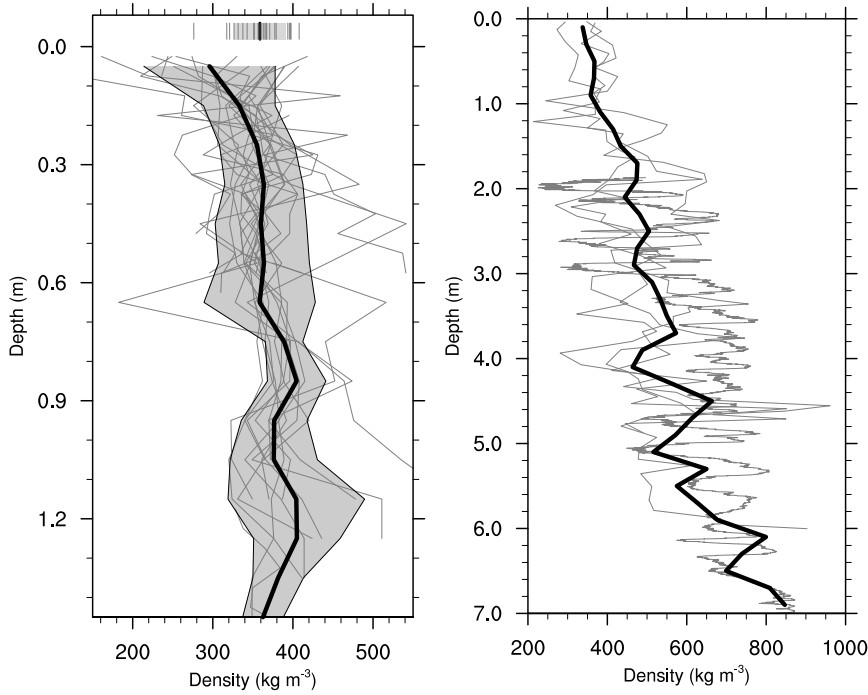

**Figure 4.** Vertical profiles of firn density used to convert the observed surface height changes to mass fluxes. Thick lines indicate mean density profiles, with shading denoting one standard deviation. (a) Snow pit observations between 0 and 1.50 m. Vertical bars in the top of the figure are vertically integrated densities, including the mean vertically-integrated density as a thick bar; (b) firn core observations (gravity coring and OPTV logging) up to 7 m below the surface.

(standard deviation at 19–44% of the mean for stations with time series longer than 3 years), but generally consistent between the different locations. For example, the winters of 2009 and 2013 stand out as high-SMB winters at all stations.

The multi-year mean winter SMB at each site is summarized in Table 2. The differences between the stations are small and mostly within error bounds. The highest mean winter SMB is seen at AWS14 at $227\pm20$ mm w.e. $y^{-1}$ for 2009–2015, and the lowest at LAR3 at $174\pm23$ mm w.e. $y^{-1}$ for 2009–2011. The winter SMB is somewhat higher near the ice-shelf margin than farther inland.

A lack of detailed density profiles at the end of the melt season precludes the construction of an annually resolved record of summer SMB (Nov–Mar) for each site. However, a multi-year mean annual SMB can be computed by using the cumulative height signal over multiple years and combining that with firn-core derived density profiles. The mean summer SMB at each site can then be calculated by subtracting the mean winter SMB from the mean annual SMB. For that, we assume that there is no summer runoff, and that the refreezing of all surface melt water happens within the annual layer of snowfall. In equations 1 and 3, this corresponds to assuming that $RU = 0$ and $m = 0$. As an estimate of the density profile required to convert height





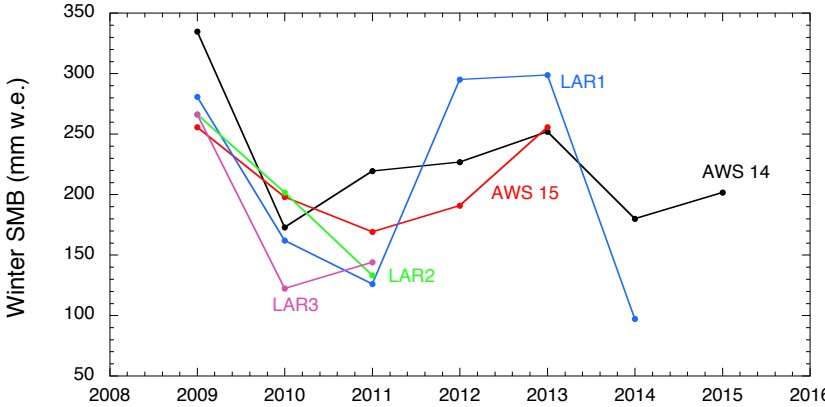

**Figure 5.** Time series of winter (Apr–Oct) surface mass balance (mm w.e.) derived from sonic height rangers.

**Table 2.** Estimates of mean winter (Apr–Oct), summer (Nov–Mar) and annual SMB (in mm w.e. $y^{-1}$) for five sites with continuous sonic height ranger observations. The rightmost column shows the period on which the estimated SMB is based. The bottom five rows show results for the longest common period of all records (Apr 2009 – Nov 2011).

| Site | Winter SMB | Summer SMB | Annual SMB | Period |
|------|-----------|-----------|-----------|--------|
| AWS14 | $227 \pm 20$ | $186 \pm 37$ | $413 \pm 42$ | Apr 09 – Oct 15 |
| AWS15 | $214 \pm 21$ | $233 \pm 36$ | $447 \pm 42$ | Apr 09 – Mar 14 |
| LAR1 | $210 \pm 21$ | $287 \pm 45$ | $496 \pm 50$ | Apr 09 – Oct 14 |
| AWS14 | $242 \pm 27$ | $194 \pm 45$ | $436 \pm 68$ | Apr 09 – Nov 11 |
| AWS15 | $208 \pm 25$ | $219 \pm 55$ | $426 \pm 59$ | Apr 09 – Nov 11 |
| LAR1 | $190 \pm 24$ | $366 \pm 63$ | $556 \pm 68$ | Apr 09 – Nov 11 |
| LAR2 | $200 \pm 25$ | $219 \pm 59$ | $419 \pm 64$ | Apr 09 – Nov 11 |
| LAR3 | $174 \pm 23$ | $221 \pm 56$ | $395 \pm 61$ | Apr 09 – Nov 11 |

change to SMB, we take the mean density profile of four available deep density profiles: three firn cores at LAR1, 2, and 3, and an OPTV density log at AWS14. This mean density profile is shown in Figure 4b.

Annual SMB (see Table 2) is highest for LAR1 ($496 \pm 50$ mm w.e. $y^{-1}$) and lowest for LAR3 ($395 \pm 61$ mm w.e. $y^{-1}$). Only the annual SMB value from LAR1 is substantially higher than at the four other locations. If we consider the common period only (2009–2011), LAR1 stands out even more at $556 \pm 68$ mm w.e. $y^{-1}$. For summer SMB (1 Nov – 31 Mar), we see that LAR1 shows the largest values by far, with ∼40% more summer mass gain than the other locations. At the other locations, the summer SMB values are almost identical at 194–221 mm w.e. $y^{-1}$.



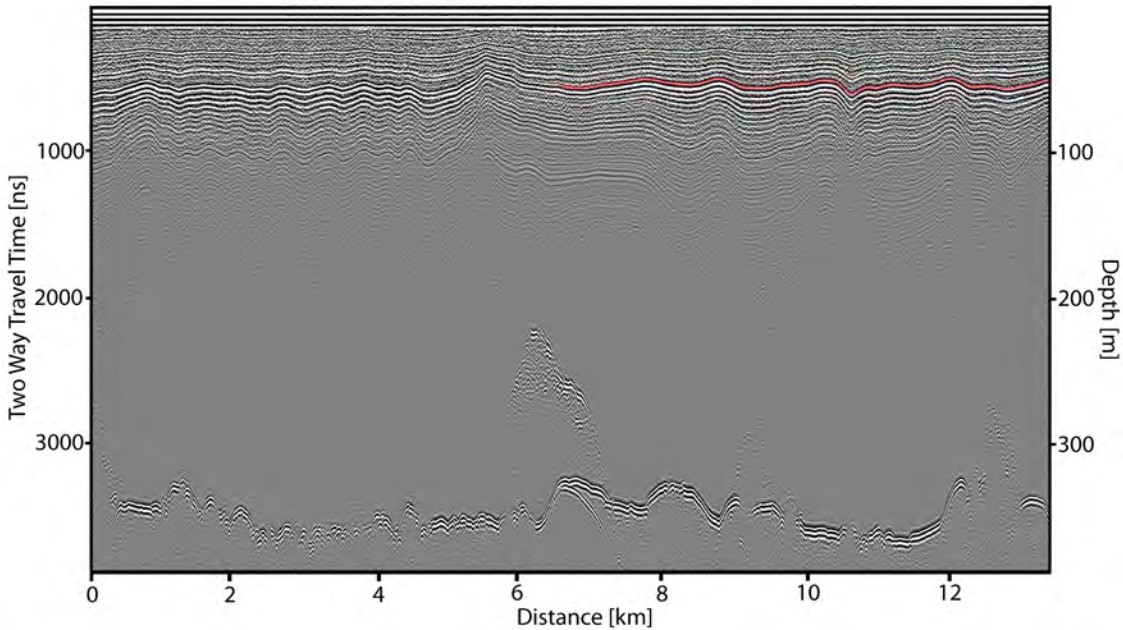

**Figure 6.** Example radargram from ground-penetrating radar, with the reflection horizon at 35–45 m below the surface clearly visible, partly indicated with a red line. The lowermost reflections are from the ice-ocean interface at about 350 m depth.

### 3.2 SMB on decadal time scales using ground-penetrating radar

In all ground-based radar surveys of LCIS, we find a particularly strong reflection horizon at 35–45 m (median 41 m) depth below the surface (see example radargram in Figure 6). Radar reflection horizons in firn and ice are related to strong contrasts in permittivity of the firn, originating e.g. from volcanic ash layers, or from changes in the firn properties, such as temperature,

5 density, fabric, and grain size. Based on the distinctiveness and spatial continuity of this layer, and lack of east-west depth gradient that would be suggestive of progressive burial with seaward ice advection, we interpret it as having formed synchronously over the ice shelf, at least in a single melt season. The strong reflection is at the same depth and of similar signature in radargrams at cross-over points. Unfortunately, the reflection horizon is undated, which precludes the conversion from accumulated mass to a rate of SMB.

10 Total accumulated mass is shown in Figure 9 for all available radar lines. The lowest values (26–28 m w.e.) are found in the northern part of LCIS along the MIDAS 2014 radar lines. The highest values (40–45 m w.e.) are concentrated near the southern end of the McGrath radar survey, and to the southwest in the SOLIS 2009 survey. The multi-decadal SMB estimates



**Table 3.** Comparison of snow height from sonic height rangers and reflector depths from OIB radar data. End-of-melt date is estimated from spaceborne scatterometry; distance denotes the minimum distance between the OIB flight track and the sonic height ranger locations.

| Name | End of melt date | OIB flight date | OIB depth (m) | SHR depth (m) | Distance (km) |
|------|------------------|-----------------|---------------|---------------|---------------|
| AWS14 | 12 Feb 2009 | 4 Nov 2009 | $1.05 \pm 0.07$ | $1.06 \pm 0.03$ | 1 |
| AWS15 | 3 Feb 2009 | 4 Nov 2009 | $1.03 \pm 0.07$ | $1.04 \pm 0.17$ | 4 |
| LAR1 | 29 Jan 2009 | 16 Nov 2009 | $1.16 \pm 0.07$ | $0.97 \pm 0.05$ | 1 |
| LAR2 | 3 Feb 2009 | 4 Nov 2009 | $1.11 \pm 0.07$ | $1.13 \pm 0.05$ | 2 |

from the radar surveys confirm the observations made by the sonic height rangers, in that the SMB is lower in the north than in the south.

A very crude way to estimate the age of the reflective layer is to divide the mean reflector depth (31.5 m) by the mean accumulation rate from the SHR observations ($0.43 \pm 0.05$ m w.e. y$^{-1}$). This gives an age of the reflector $73 \pm 8$ y, putting the
5  year of origin of this layer in the 1930s or 1940s.

### 3.3 Airborne radar and spatial wintertime snowfall

Here, we present airborne radar as another, independent method capable of mapping spatial variability in winter SMB across the ice shelf. The OIB Ku-band shows a spatially persistent, single reflection horizon approximately 0.70 to 1.40 m below the surface, across the entire ice shelf (Figure 7). The presence of one single strong reflector is anomalous compared to data
collected over other parts of Antarctica, where multiple reflectors provide information on the vertical layering in the top few meters of the firn (e.g. Medley et al., 2015). The OIB radar data are unlike to low-frequency GPR data sets, in that there is only a single strong subsurface horizon. This is likely related to differing scattering characteristics of the firn at the much higher OIB frequency. We hypothesize that the single reflection horizon across LCIS coincides with the top of the melt layer formed during the previous austral summer.
To test this hypothesis, we compared the OIB reflector depth with the four sonic height rangers for which data are available in 2009. Specifically, we looked at the thickness of the snow layer that had accumulated since the last significant melt event of the melt season preceding the OIB flights. The melt-season termination dates were established using QuikSCAT and ASCAT microwave data (Trusel et al., 2013). Uncertainty in the sonic height ranger data is computed based on instrument measurement error, and uncertainty in the melt termination date derived from the microwave satellite data. In most cases, this amounts to a
few cm, depending on the snow height variability around the melt season termination date. At AWS15 in 2009 however, the approximated end-of-melt date coincides with a major snowfall event, raising the uncertainty in the sonic height ranger data to 0.17 m. Uncertainty in the OIB reflector depth is estimated at 0.07 m (see Methods section). The comparison, compiled in Table 3, shows that, at three locations, there is agreement within 0.02 m between the observed snow accumulation since the previous melt season, and the depth of the OIB-observed reflection horizon. There is a discrepancy between OIB-estimated



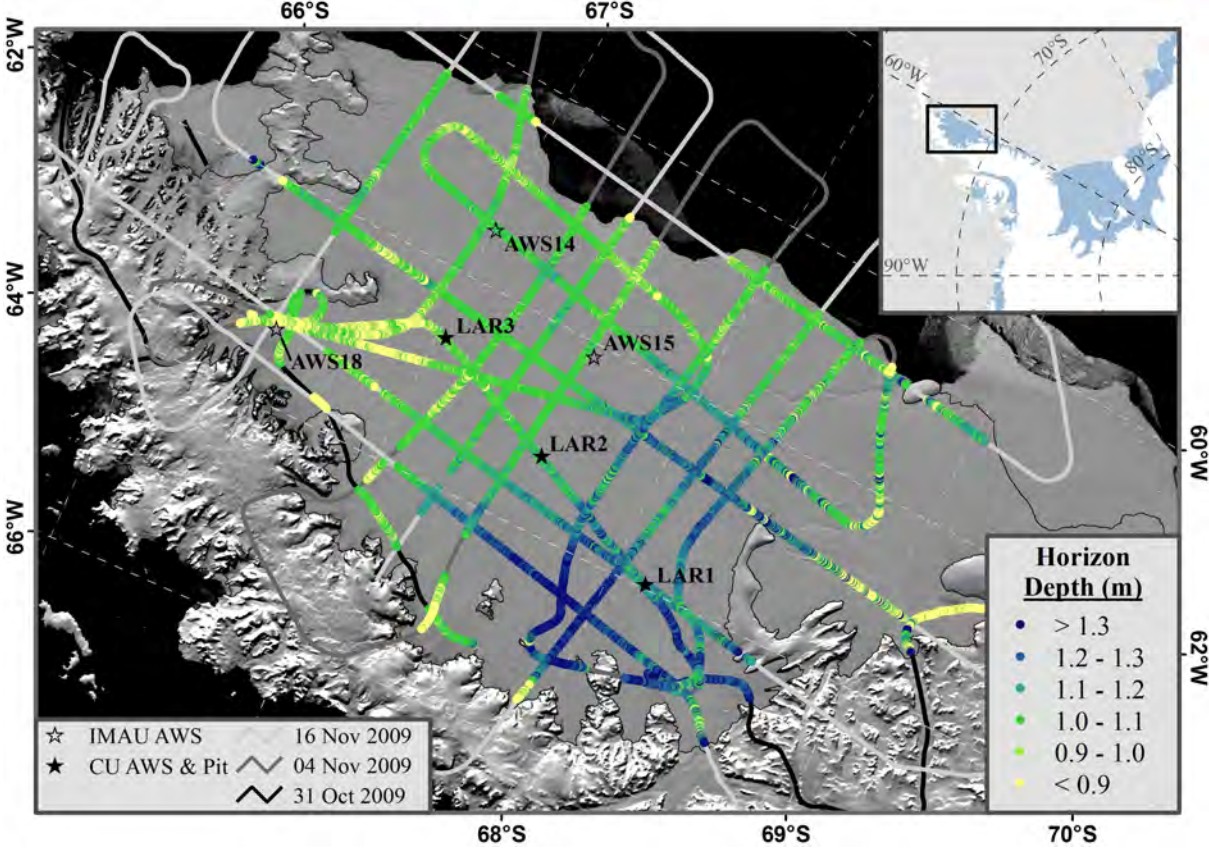

**Figure 7.** Map of LCIS showing Operation Ice Bridge reflector depths (in m) for the 2009 flight lines. Stars indicate the locations of automatic weather stations, with solid stars at locations where snow pit observations were made concurrent with the OIB overpasses.

and SHR depth at LAR1 that largely exceeds the uncertainty estimates. The LAR1 SHR appears out of pattern but we could not establish the cause for this discrepancy.

In Figure 8, we compare OIB-derived accumulation for the period Feb-Oct 2009 with RACMO2-simulated SMB for the same period (assuming again a density of 360 kg m$^{-3}$). At a total of 415 equally-spaced locations along the OIB flight
5     lines, the mean bias is 0.10 m (RACMO2: 0.97 m, and OIB: 1.07 m). RACMO2 seems to underestimate at low-accumulation locations, and to overestimate at high-accumulation locations, leading to a slope that is larger, but not statistically significantly larger, than one. Nonetheless, RACMO2 seems to capture the magnitude and the spatial pattern of accumulation for this winter season.

Having established the OIB reflection horizon over LCIS to approximate snowfall since the last snowmelt event of the
10     previous melt season, we are able to map the spatial variability of snow accumulation for the austral winter of 2009 at locations where OIB data are available (Figure 7). We see diminished winter accumulation in the northern part of the ice shelf, with



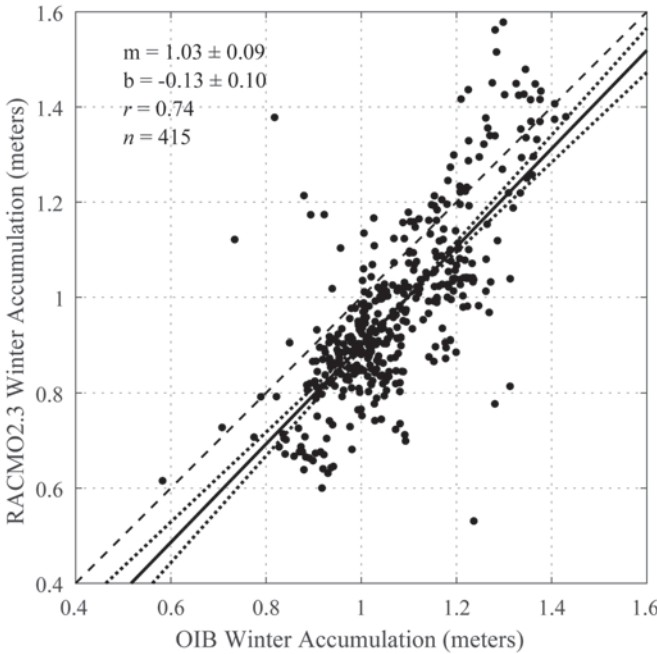

**Figure 8.** Comparison between Operation Ice Bridge-derived and RACMO2-simulated winter accumulation (m) between February and November 2009, based on 415 equally-spaced observations along the OIB flight lines. The 1:1-line is dashed; the linear fit is represented by a solid line, with associated uncertainty of the fit in dashed lines. Slope of the fit line is $m$ and intercept with the vertical axis is $b$.

smallest values in the northwestern inlets of LCIS. Higher accumulation is found in the southern part, with particularly high values around LAR1 in the southwestern part of the ice shelf. In the far south of the ice shelf (south of the Kenyon Peninsula), we see again much lower amounts of accumulation.

### 3.4 A map of relative SMB and its origin

At all timescales examined in this study – from seasonal to multidecadal – we find that higher SMB values are found in the middle and southern sectors of LCIS, and lower SMB in the north. In order to expand our coverage to unsurveyed areas of LCIS, we use the regional climate model RACMO2 (see Methods section). Underestimation of RACMO2 snowfall over LCIS was noted by Kuipers Munneke et al. (2014a) and may be the result of the representation of snow formation in clouds, or with underestimated evaporation in the Weddell Sea, the most important source region for moisture precipitated over LCIS. Due to this underestimation, we use relative rather than absolute values of SMB. We average RACMO2 SMB over 1979–2014, and normalize it with respect to the spatially mean SMB over the ice shelf. The GPR data is another source of relative, but not absolute, rates of multi-decadal SMB, which we also normalize with respect to its spatial mean. Next, we determine a linear regression of the normalized RACMO2 SMB values to the normalized GPR data. We use this regression to adjust the RACMO2 SMB data to maximize its match to the GPR data. The result is a RACMO2-guided extrapolation of the GPR over





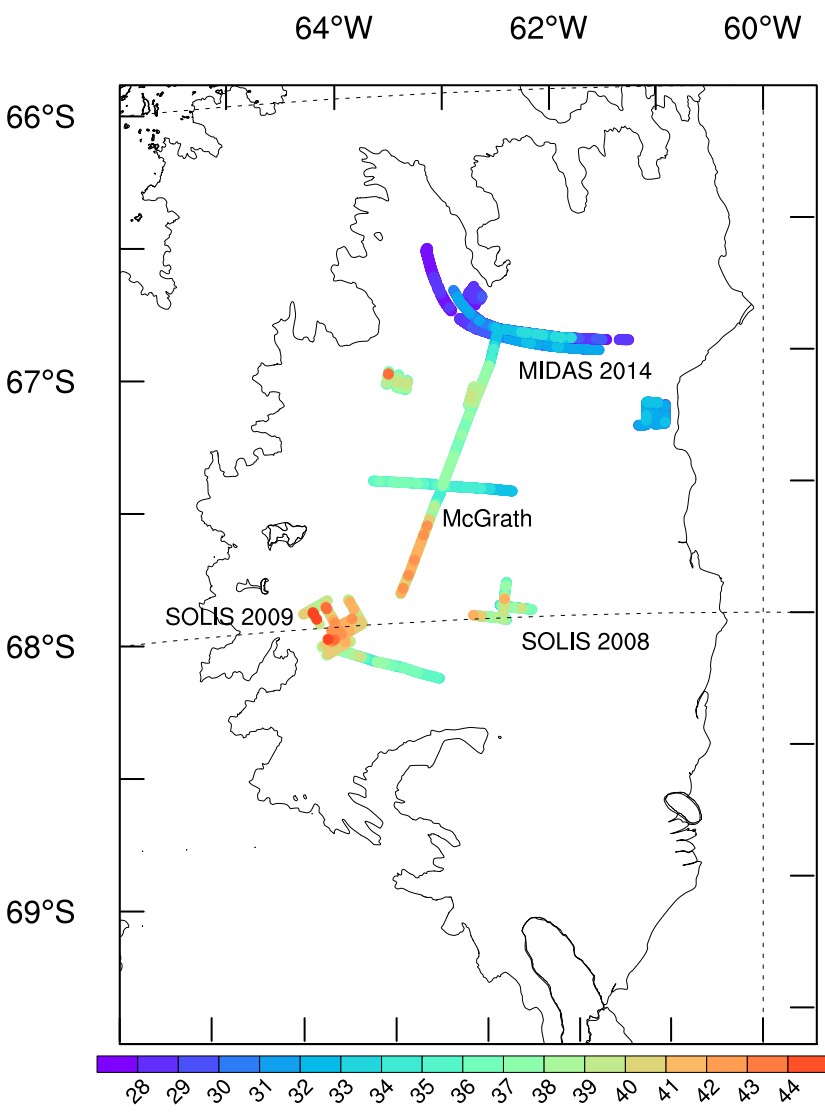

**Figure 9.** Map of LCIS showing estimated total mass (m w.e.) above the strong, undated reflection horizon along the GPR tracks.





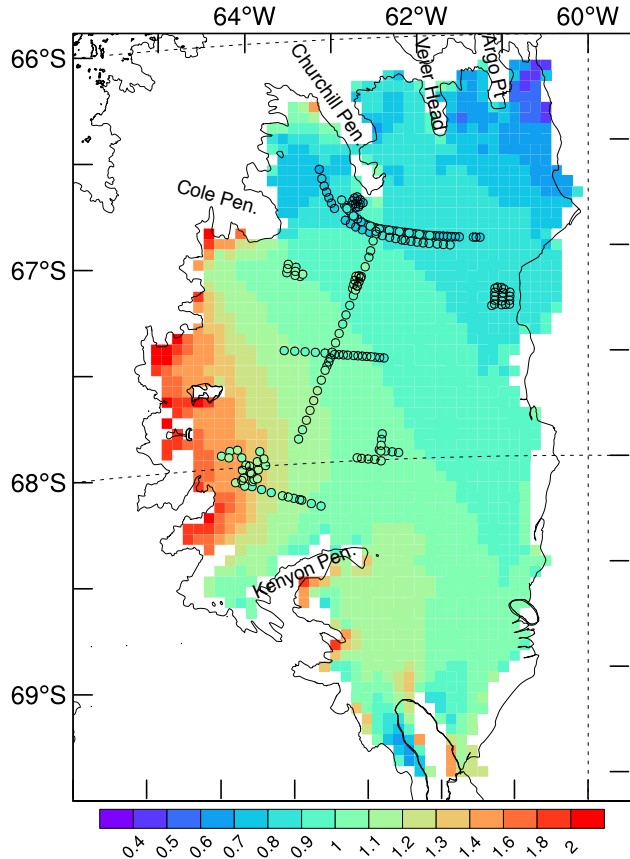

**Figure 10.** Coloured dots indicate depth of the reflection horizon, detected by GPR, normalized with respect to the mean depth (unitless). The background shows a map of relative SMB (unitless) from RACMO2.

the unsurveyed portions of LCIS, shown in Figure 10. It shows broader context to the different observations presented above. In the area of GPR observations, the RACMO2-guided interpolation shows that the SMB gradient is not strictly north-south, but tilted in the northeast-southwest direction, with lowest values near Bawden Ice Rise in the northeast, and highest values of SMB in the inlets in the west-to-southwest part of the ice shelf.

5    We use RACMO2 to study the origin of this spatial distribution of SMB. In the absence of notable runoff, SMB is dictated by snowfall. The map in Figure 11a illustrates the spatial coherence of snowfall across LCIS, by showing the fraction of snowfall occurring simultaneously with snowfall events exceeding 5 mm w.e.d$^{-1}$ in the southern area of LCIS (results are relatively insensitive to the exact location on LCIS). The pattern shows that snowfall is very coherent across the shelf, and that only a small fraction of snowfall on the western side of the Peninsula occurs when there is snowfall on its eastern side. Figure 11b

10   shows the mean circulation pattern during these snowfall events (wind speed and direction at the 850 hPa pressure level, along with temperature at 850 hPa in the background). Snowfall on LCIS is thus strongly associated with low pressure centered near

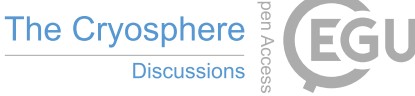



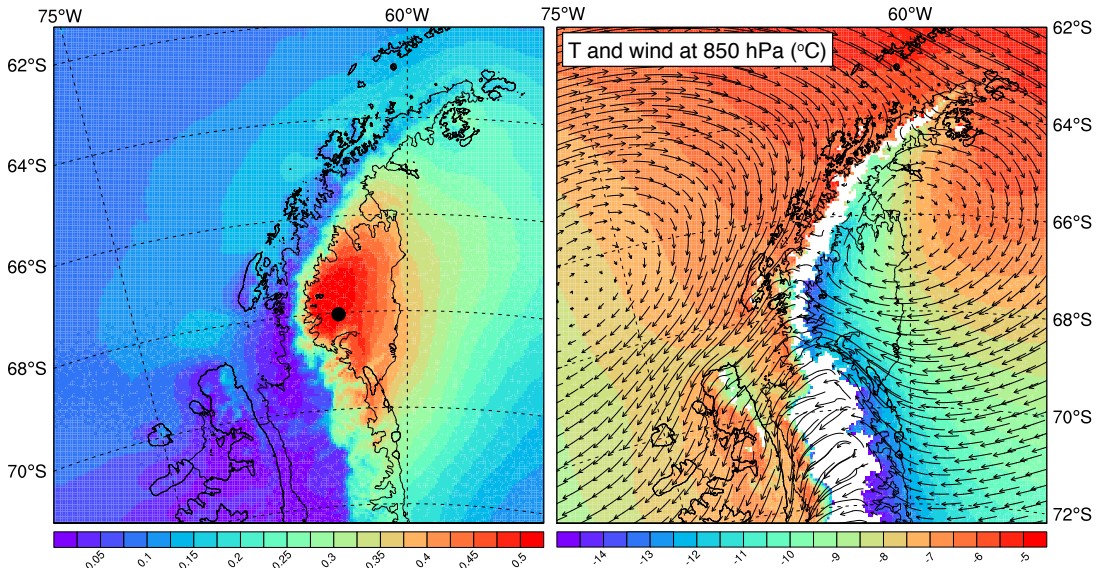

**Figure 11.** Maps showing (a) the fraction of annual snowfall on days when snowfall on southern LCIS (location given by black dot in the left panel) exceeds 5 mm w.e. d$^{-1}$; and (b) the mean wind speed and direction, and air temperature on those days.

its northern end over the Weddell Sea. These low pressure systems source water vapor from the Weddell Sea and from more northerly regions to produce snowfall on LCIS. This circulation pattern can explain the relatively high snowfall rates in the south and southwest, where snowfall is orographically enhanced. As a peculiar smaller-scale expression of this orographic effect, the southeastern side of the Kenyon Peninsula receives more snowfall than its northwestern side, situated in the lee

of the flow associated with snowfall. The same pattern can be seen around the promontories clockwise around the ice shelf (Cole and Churchill Peninsulas, Veier Head, and Argo Point), where reduction of snowfall is seen at the obstacle's lee side. The snowfall minimum in the north can be explained by the fact that the low-pressure center will fluctuate around the mean center in Figure 11b. If the low-pressure system is located farther to the south, the northeastern part of LCIS will experience an off-shore wind, in the lee of the Jason Peninsula and its promontories extending to the south (Churchill Peninsula, Veier Head,

and Argo Point). Such a southerly position of the low-pressure system would not influence snowfall on much of the ice shelf, except for the northeastern part. Thus, snowfall in the northeast of LCIS is more restricted during local off-shore wind than in other places.

## 4   Conclusions

We have combined several geophysical techniques along with a regional climate model to constrain spatial and temporal

patterns of SMB over the Larsen C Ice Shelf. Results have been integrated to show that SMB is larger towards the south of the ice shelf, overprinted by an increase of SMB toward the west. Assuming that runoff is negligibly small over LCIS (Van





Wessem et al., 2016), the spatial pattern of SMB is dominated by spatial differences in snowfall. Thus, our results indicate that snowfall is larger in the south than in the north of LCIS.

Previous studies have indicated a strong gradient in firn air content from west to east across LCIS (Holland et al., 2011), with the lowest values in the west. It has been suggested that this reflects enhanced melt and subsequent refreezing, directly at the foot of the Antarctic Peninsula mountains in the western part of LCIS (Trusel et al., 2013) caused by föhn winds descending from the mountains (Luckman et al., 2014). Using observations of SMB, this study shows an east-to-west SMB gradient that, in the absence of melt, would lead to the highest, rather than the lowest, values of firn air in the west. We therefore conclude that the gradient in firn air content is caused by melt and refreezing, rather than by spatial patterns of snowfall or SMB.

We interpret a strong, shallow reflection horizon in the OIB radar data as the top of the melt layer formed during the previous melt season. The presence of sufficiently thick melt layers, like on LCIS, precludes airborne observations of multi-year firn stratigraphy like in dry areas of Antarctica and Greenland. Still, we demonstrate that OIB radar data can be used to track melt layers and winter SMB. This opens up the possibility of acquiring winter SMB estimates over other Antarctic ice shelves, and over parts of the Greenland Ice Sheet that experience small to moderate amounts of melt, as is typical for percolation zones.

Much recent work has focused on the stability of LCIS in a warming climate, with hydrofracturing suggested as a potential mechanism for ice-shelf collapse (Scambos et al., 2000; Kuipers Munneke et al., 2014b; DeConto and Pollard, 2016). Using models to test hypotheses that link atmospheric change to ice-shelf stability is challenging, given the complexity of terrain and climate in this region. A substantial portion of total melt is governed by the occurrence of small-scale föhn winds, that are captured reliably only in models with km-scale horizontal resolution (Elvidge et al., 2014). Summertime melt is also observed frequently on other days (Kuipers Munneke et al., 2012), and its representation in models depends on subtle changes in albedo (Kuipers Munneke et al., 2011), and on the correct simulation of all components of the radiation balance at the surface (King et al., 2015).

Precipitation depends on the ability of existing weather fronts to cross the Antarctic Peninsula mountain range, but also on local lee cyclogenesis, and on low-pressure systems crossing the Weddell Sea. The moisture content of the latter depends on both sea ice extent and the presence of polynyas. Thus, LCIS is an important and challenging testbed for regional, global, and Earth system models. Performance of these models to predict ice-shelf collapse is subject to a correct representation of observed climate. Estimates of SMB presented in this study can guide model evaluation and development with the aim of improving our capacity to predict stability of LCIS.

*Author contributions.* P. K. M., D. M. and B. M. conceived this study, and performed the analysis and synthesis of the data sets. P. K. M. led the writing of the manuscript. D. M., B. M., S. B., B. K., D. J., A. B., P. S., D. A., A. S., and N. G. processed and provided observational data sets. A. L., S. B., B. K., A. B., B. H., D. A., and H. S. collected data in two MIDAS fieldwork campaigns. All authors contributed to discussions in writing this manuscript.

*Competing interests.* The authors declare that they have no conflict of interest.



*Acknowledgements.* This work is funded by the Netherlands Polar Programme, Netherlands Earth System Science Centre (NESSC), NSF OPP research grant 0732946, NERC/GEF grants NE/L006707/1, NE/L005409/1, NE/E012914/1, GEF loans 863, 890, 1028. We thank logistical support from the British Antarctic Survey during the various field campaigns. We also acknowledge the generous contribution of faculty, staff, and students at CReSIS in collecting and processing the Ku-band data as well as NASA's Operation IceBridge team in collecting and disseminating data to the public. Use of trade, product, or firm names is for descriptive purposes only and does not imply endorsement by the U.S. Government.



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
