# Peer review of "Observationally constrained surface mass balance of Larsen C Ice Shelf, Antarctica"

_The Cryosphere, 2017_

## Referee Comment (RC1) · Anonymous Referee #1 · 7 May 2017

Well-written and thoroughly described set of observations, and comparison with climate models, discussing the accumulation pattern and SMB variations of the Larsen C ice shelf. Overall, an excellent compilation of work by several groups that addresses one issue, and integrates the variety of observations with a model to come up with a consistent picture. While nothing too surprising is noted, the number of measurements, breadth of measurements, careful corrections, etc. make this a highly citable reference for Larsen C net surface mass balance in the early 21st century (and extending earlier, through the radar profile measurements).

However, the Discussion and Conclusions need to be expanded significantly to include foehn effects – these are events that occur without concurrent snowfall, that do indeed lead to strong surface melting as you note, but also result in high rates of evaporation, and therefore a reduction in the net SMB. Seasonally and annually. Clearly the RACMO

model needs to improve this aspect of its estimation of conditions in the area. Foehn events can occur at any time of year in the northern Larsen C. A component of the explanation for the observed northeast-to-southwest gradient is a north (high) to south (lower) gradient in foehn frequency along the western margin of the LCIS, and therefore in the evaporation component. Figure 10 is making this point quite clear, as is Figure 7. Foehn effects are localized, but significant. Previous studies of SMB on the northern Peninsula (van Wessem et al., 2016) have made it clear that the 'snow desert' in the lee of the Peninsula (i.e. the eastern ice shelves) is due to foehn.

Abstract •Abstract, and throughout the paper — Why not use cm for the SMB values, rather than mm? since that is a more appropriate scale to use given the error and nature of the measured quantity? •Last sentences – but you show that SMB does in fact decrease to the west, due to the ablation effects of foehn wind — and this has also been highlighted recently in Cape et al., 2014; Turner et al., 2016, Oliva et al, 2017. You should look at / include these papers in the Discussion. The end of the abstract needs to be re-written to better reflect all your results.

Introduction •you cite Turner et al. , 2016 here but not in references; also, not sure which paper you are referring to.

P4L23 – remove 'in reality'... it's all reality. P6L15 – untangle epsilon and the reference: "...its dielectric constant (e; Kovacs et al., 1995), which..." P10L10 – change to: 'This assumes that there is no...' passive voice is actually clearer here. Table 2 – The errors in Winter SMB seem too small, given Figure 5. P12L4 – although it is true that the permittivity would change with temperature, by listing it first you imply that it would be the most common, and that's not the case, the other three are more commonly invoked (if not volcanic ash/acid). P13L5 – I wonder if Alison Cook (who I think has looked at every aerial photography taken in the pre-satellite era) would have a notion about a year of extensive melting during this 1930s1940s period... just a thought. P13L6 – I would start this as 'Airborne radar is another, independent....' P13L11 – check wording and meaning here: '...radar data are unlikely to low-frequency...' P13L13-14 –

you already introduced this idea in methods with Figure 2. . . need to re-state somehow, further up or here. Perhaps the last line of the paragraph here could be deleted, and the first line of the next paragraph could begin: "To test our earlier assumption that this reflector is the previous summer's melt horizon, . . ." P13L22 – In methods, the error was given as 6cm? (based on 2GHz bandwidth) ..reconcile. P17L1 and 2 – RACMO-2 is a model. . . While it 'provides a broader context to the different observations presented above' (suggested wording change) it really can only '. . .suggest (or imply)' that the SMB gradient is northeast-southwest (suggested wording change). P17L5-6 – No, it is dictated by snowfall and evaporation – and RACMO2 is clearly not getting the foehn-derived evaporation component right.

Figure 1 – please use the data citation for Mosaic of Antarctica (Haran et al., 2014. . . at nsidc.org) ; and the data citation for the IceBridge flightlines as well. Figure 2 – Need to describe the sensor used for this radar profile in the caption, and, could indicate where this profile came from in Figure 1 (latter not all that critical) Figure 4 – you could make the depth-integrated lines at the top of (a) bigger – push the 0.0 depth scale down a bit to make more room. . .. Nice plot over all. Figure 5 – this plot would take on additional meaning if you added the mean SAM index for the winter period as a bar graph along the bottom, with a right-side y-axis. Start the left-side y-axis at 0 to make room. This would warrant a paragraph discussion in the Discussion. Figure 6 – please note where on the Larsen C this profile was acquired, perhaps in Figure 1. This one is a bit more important, since it shows some location – specific structure at depth.

---

## Referee Comment (RC2) · Anonymous Referee #2 · 26 May 2017

This paper goes to a lot of trouble to explain how the different data sources with their various strengths and weaknesses are used to estimate surface mass balance (SMB) over Larsen C Ice Shelf (LCIS). There are other scientists who can assess this part of the work much better than I can. From a practical point of view however, the qualitative conclusion that SMB increases from north to south overprinted with a gradient of increasing SMB to the west is a major disappointment and fails the stated aim on line 20 to provide a coherent picture of SMB for LCIS. Surely the goal should be a grid of SMB values in mm of water equivalent for a particular set of years, and even better broken into winter and summer. The paper is on the verge of doing this but doesn't deliver. The authors should perform such an analysis as part of this manuscript.

---

## Author Comment (AC1) · 5 Jul 2017

**Response to reviewer #1**

*[...] [T]he Discussion and Conclusions need to be expanded significantly to include foehn effects – these are events that occur without concurrent snowfall, that do indeed lead to strong surface melting as you note, but also result in high rates of evaporation, and therefore a reduction in the net SMB. Seasonally and annually. Clearly the RACMO model needs to improve this aspect of its estimation of conditions in the area. Foehn events can occur at any time of year in the northern Larsen C. A component of the explanation for the observed northeast-to-southwest gradient is a north (high) to south (lower) gradient in foehn frequency along the western margin of the LCIS, and therefore in the evaporation component. Figure 10 is making this point quite clear, as is Figure 7. Foehn effects are localized, but significant. Previous studies of SMB on the northern Peninsula (van Wessem et al., 2016) have made it clear that the 'snow desert' in the lee of the Peninsula (i.e. the eastern ice shelves) is due to foehn.*

We acknowledge that sublimation by foehn winds has a modulating effect on Larsen C SMB patterns, with the largest effect in the inlets along the western edge of the ice shelf. The sublimation flux is calculated as part of the RACMO2 SMB fluxes, and below, we try to argue that the sublimation fluxes as simulated by RACMO2 are in agreement with in situ observations.

[Figure]

In a forthcoming paper, we will present sublimation fluxes from an automatic weather station located in Cabinet Inlet (see right panel above). Because the relative humidity sensor exhibited some problems, the estimate of sublimation at this location have greater uncertainly, but range from 17 to 64 mm w.e. or 5 to 18% of the annual surface mass balance. RACMO2 estimates sublimation at this location to be 42 mm w.e. (left panel).

For another location (AWS 14), the mean sublimation flux computed from the weather station observations is ~32 mm w.e. RACMO2 simulates a mean flux between 25 and 30 mm w.e. (left panel).

RACMO2 shows patterns of enhanced sublimation, in places where foehn occurs (left panel); and reduced sublimation downstream of prominent peninsulas extending into the shelf. The differences in sublimation can be large (a factor 3) over short distances, particularly in and around the inlets. This leads us to believe that RACMO2 does a reasonable job at simulating sublimation, including during foehn conditions, in different places of the ice sheet. A sublimation flux of 5-15% of the SMB is in line with observations from other foehn or katabatic wind sites (e.g., Van den Broeke et al., 2010, Antarctic Science).

We acknowledge that sublimation by foehn winds modulates the SMB on the western part of the shelf. This is discussed in the revised manuscript. We feel confident in using RACMO2-simulated fluxes (we acknowledge that it is merely a model) to illustrate this modulating effect on SMB. This modulating effect is now mentioned in an additional paragraph in section 3.4, and in the abstract:

**Sublimation exerts a secondary control over SMB. Over LCIS, föhn winds are frequent, and the combination of high wind speed and dry air increases the sublimation rate. During föhn, a pattern of alternating higher and lower wind speed emerges in the western part of LCIS, where low-elevation inlets are separated by higher-elevation promontories that protrude from the Antarctic Peninsula mountains (Luckman et al, 2014; Elvidge et al., 2015). An estimate of annual mean sublimation rate from RACMO2 is shown in Figure 12.**

While sublimation rates from RACMO2 are poorly evaluated over LCIS, an estimate of sublimation from *in situ* AWS observations reveals that it amounts to ~25-30 mm w.e. $y^{-1}$ at AWS 14 (see map in Figure 1), and between 17 and 64 mm w.e. $y^{-1}$ at the site of a newly installed AWS in Cabinet Inlet. Comparing the sublimation flux (Figure 12) to the total SMB (Figure 10) shows that sublimation spatially modulates SMB in the western part of LCIS, likely by föhn. According to RACMO2, annual mean sublimation typically removes 5-15% of the annual snowfall over LCIS. This fraction could be larger if RACMO2 underestimates the sublimation flux. It is conceivable that the SMB in the inlets (Cabinet, Mill, Whirlwind, Mobiloil) has decreased in recent decades following intensification of föhn (Cape et al., 2016), due to enhanced sublimation.

In the abstract, we added: Combining snow height observations, ground and airborne radar with SMB output from a regional climate model yields a gridded estimate of SMB over LCIS. It confirms that SMB increases from north to south, overprinted by a gradient of increasing SMB to the west, modulated in the west by föhn-induced sublimation.

*Abstract, and throughout the paper — Why not use cm for the SMB values, rather than mm? since that is a more appropriate scale to use given the error and nature of the measured quantity?*

We have tried to reconcile this by stating all SMB values in meters w.e. For the results of our study, we use a two-decimal accuracy (e.g., 0.43 m w.e. rather than 428 mm w.e.). Only where more accuracy is given in existing literature, we add an additional decimal (e.g. 0.390 and 0.404 m w.e. $y^{-1}$ for the Dolleman ice cores in section 1).

*Last sentences – but you show that SMB does in fact decrease to the west, due to the ablation effects of foehn wind — and this has also been highlighted recently in Cape et al., 2014; Turner et al., 2016, Oliva et al, 2017. You should look at / include these papers in the Discussion. The end of the abstract needs to be re-written to better reflect all your results.*

The RACMO2 SMB suggests an increase to the west, apart from the inlets where SMB remains constant or decreases somewhat to the west. This is indeed caused by the modulating effect of foehn and is now included in the manuscript, also in the end of the abstract (see comments above).

*you cite Turner et al. , 2016 here but not in references; also, not sure which paper you are referring to.*

In fact, Turner et al. (2016) is in the reference list: Absence of 21st century warming on Antarctic Peninsula consistent with natural variability, Nature, 535, 411-415.

*P4L23 – remove 'in reality'... it's all reality.*

Removed and changed to: The sonic height rangers themselves were always located between 1 and 4 m above the surface.

*P6L15 – untangle epsilon and the reference: "... its dielectric constant (e; Kovacs et al., 1995), which ..."*

This was indeed unclear. Changed.

*P10L10 – change to: 'This assumes that there is no ...' passive voice is actually clearer here.*

Adjusted.

*Table 2 – The errors in Winter SMB seem too small, given Figure 5.*

We have clarified that the stated errors in Table 2 reflect measurement uncertainty, but not the interannual variability that is shown in Figure 5. The uncertainty estimate is valid for the cumulative winter, summer, and annual SMB for the multi-year periods indicated in the rightmost column. In the caption, we added: The confidence interval reflects the measurement accuracy, not the interannual variability. In the text in section

3.1, we added: **The multi-year mean winter SMB at each site is summarized in Table 2, with confidence intervals reflecting measurement accuracy, not interannual variability in Figure 5.**

*P12L4 – although it is true that the permittivity would change with temperature, by listing it first you imply that it would be the most common, and that's not the case, the other three are more commonly invoked (if not volcanic ash/acid).*

We have moved temperature to the end of the sentence: **... or from changes in the firn properties, such as density, fabric, grain size, and temperature.**

*P13L5 – I wonder if Alison Cook (who I think has looked at every aerial photography taken in the pre-satellite era) would have a notion about a year of extensive melting during this 1930s1940s period*
*... just a thought.*

*P13L6 – I would start this as 'Airborne radar is another, independent ....'*

Changed.

*P13L11 – check wording and meaning here: ' ... radar data are unlike to low-frequency .'*

We have rephrased this to: **Unlike low-frequency GPR, the OIB radar data only show a single strong subsurface horizon. This is likely related ...**

*P13L13-14 – you already introduced this idea in methods with Figure 2 ... need to re-state somehow, further up or here. Perhaps the last line of the paragraph here could be deleted, and the first line of the next paragraph could begin: "To test our earlier assumption that this reflector is the previous summer's melt horizon, ..."*

We followed the reviewer here and added a reference to the hypothesis introduced in section 2.5: **To test our earlier assumption that this horizon represents the top of the melt layer formed during the previous melt season (section 2.5), we compared the OIB reflector ...**

*P13L22 – In methods, the error was given as 6cm? (based on 2GHz bandwidth) ..reconcile.*

The typical error is 6 cm as stated in ...

*P17L1 and 2 – RACMO-2 is a model ... While it 'provides a broader context to the different observations presented above' (suggested wording change) it really can only ' ... suggest (or imply)' that the SMB gradient is northeast-southwest (suggested wording change).*

We have adapted this suggested wording change: **The SMB pattern in Figure 10a provides a broader context to the various data sets presented above. In the area of GPR observations, the RACMO2-guided interpolation suggest that the SMB gradient ...**

*P17L5-6 – No, it is dictated by snowfall and evaporation – and RACMO2 is clearly not getting the foehn-derived evaporation component right.*

As discussed above, we acknowledge that the sublimation flux from RACMO2 is poorly evaluated, but the limited amount of in-situ observations support the order-of-magnitude estimates of RACMO2. We have added the effect of sublimation here: **We use RACMO2 to study the origin of this spatial distribution of SMB. In the absence of notable runoff, SMB is dictated by snowfall, and by sublimation.**

*Figure 1 – please use the data citation for Mosaic of Antarctica (Haran et al., 2014 ... at nsidc.org) ; and the data citation for the IceBridge flightlines as well.*

We have added these references.

*Figure 2 – Need to describe the sensor used for this radar profile in the caption, and, could indicate where this profile came from in Figure 1 (latter not all that critical)*

In the caption, we modified: **Sample radar echogram obtained from CReSIS Ku-band radar (14-16 GHz) onboard the 16 November 2009 Operation Ice Bridge flight. Automated surface and subsurface picks are overlaid.**

*Figure 4 – you could make the depth-integrated lines at the top of (a) bigger – push the 0.0 depth scale down a bit to make more room .... Nice plot over all.*

We have increased the height of the depth-integrated lines by a factor 2.5.

*Figure 5 – this plot would take on additional meaning if you added the mean SAM index for the winter period as a bar graph along the bottom, with a right-side y-axis. Start the left-side y-axis at 0 to make room. This would warrant a paragraph discussion in the Discussion.*

We looked at winter SAM but found very low correlations with any of the stations. We added SAM in figure 5, and we added: **In this timespan, we found no link between SMB and the southern annular mode (SAM, shown as gray bars in Figure 5), with values of $R^2$ lower than 0.3. Apparently, SAM is not a good indicator for the occurrence of precipitation, like it is for temperature and summer melt, due to enhanced föhn during negative SAM (Cape et al., 2016).**

*Figure 6 – please note where on the Larsen C this profile was acquired, perhaps in Figure 1. This one is a bit more important, since it shows some location – specific structure at depth.*

This has been added to Figure 1.

---

## Author Comment (AC2) · 5 Jul 2017

**Response to reviewer #2**

*This paper goes to a lot of trouble to explain how the different data sources with their various strengths and weaknesses are used to estimate surface mass balance (SMB) over Larsen C Ice Shelf (LCIS). There are other scientists who can assess this part of the work much better than I can. From a practical point of view however, the qualitative conclusion that SMB increases from north to south overprinted with a gradient of increasing SMB to the west is a major disappointment and fails the stated aim on line 20 to provide a coherent picture of SMB for LCIS. Surely the goal should be a grid of SMB values in mm of water equivalent for a particular set of years, and even better broken into winter and summer. The paper is on the verge of doing this but doesn't deliver. The authors should perform such an analysis as part of this manuscript.*

In the original manuscript, we were reluctant to provide a gridded SMB product, and instead presented a map of gridded normalized SMB in figure 10. However, motivated by reviewer #2, we have looked into a way to connect three sources of information to construct an estimate of absolute SMB values.

The first step is already in the manuscript: the pattern of normalized snow mass above the reflection horizon is used to adjust the spatial pattern of SMB from RACMO2. Added in the revised manuscript is the next step, in which RACMO2 SMB is adjusted to match the sonic height ranger observations. This process is described in the fully revised section 3.4 (which now has the title A map of SMB and its origin):

The 1979-2014 average SMB from RACMO2 was normalized with respect to its spatial mean, and so were the GPR data. Next, we determined a linear regression of the normalized RACMO2 SMB values to the normalized GPR data. We used this regression to adjust the RACMO2 SMB to maximize its match to the GPR data while conserving the spatial mean SMB. The result is a RACMO2-guided extrapolation of the GPR over the unsurveyed portions of LCIS, shown in Figure 10c, where the spatial pattern of RACMO2 SMB is adjusted to the spatial pattern of the GPR observations.

The next step was to adjust the absolute values of RACMO2 SMB to available sonic height ranger observations. We converted RACMO2 SMB back from normalized to absolute values, again using the spatial mean SMB. We determined a weighted mean bias between RACMO2 SMB and all available sonic height ranger observations, selecting the periods for which both were available. We used the length of the height ranger observation period as a weight for the averaging, reflecting that short-term variability plays a smaller role in longer time series. Compared to the sonic height rangers, RACMO2 underestimated SMB by 14 ± 10%. Applying a bias adjustment leads to the gridded SMB shown in Figure 10a. An estimate of SMB uncertainty was based on (1) the fit between normalized GPR and RACMO2 SMB, and on (2) the 10% uncertainty of the RACMO2 bias. The resulting uncertainty is typically 15% of the SMB value, shown in Figure 10b.

The underestimation of RACMO2 snowfall over LCIS was noted by Kuipers Munneke et al. (2014) and may be the result of the representation of snow formation in clouds, or with underestimated evaporation in the Weddell Sea, the most important source region for moisture precipitated over LCIS. The underestimation of RACMO2 snowfall is also apparent in the comparison with Operation Ice Bridge radar data and amounts to -13 ± 10% (Figure 8), reinforcing the robustness of our bias estimate.